# Molecular Regulation of Androgen Receptors in Major Female Reproductive System Cancers

**DOI:** 10.3390/ijms23147556

**Published:** 2022-07-08

**Authors:** Sujun Wu, Kun Yu, Zhengxing Lian, Shoulong Deng

**Affiliations:** 1College of Animal Science and Technology, China Agricultural University, Beijing 100193, China; amywsj@cau.edu.cn (S.W.); yukun@cau.edu.cn (K.Y.); 2College of Animal Science, Shanxi Agricultural University, Jinzhong 030801, China; 3National Health Commission (NHC) of China Key Laboratory of Human Disease Comparative Medicine, Institute of Laboratory Animal Sciences, Chinese Academy of Medical Sciences and Comparative Medicine Center, Peking Union Medical College, Beijing 100021, China

**Keywords:** androgen, androgen receptor, ovarian cancer, cervical cancer, endometrial carcinoma

## Abstract

There are three main types of cancer in the female reproductive system, specifically ovarian cancer (OVCA), endometrial cancer (EC), and cervical cancer (CC). They are common malignant tumors in women worldwide, with high morbidity and mortality. In recent years, androgen receptors (ARs) have been found to be closely related to the occurrence, progression, prognosis, and drug resistance of these three types of tumors. This paper summarizes current views on the role of AR in female reproductive system cancer, the associations between female reproductive system cancers and AR expression and polymorphisms. AR regulates the downstream target genes transcriptional activity and the expression via interacting with coactivators/corepressors and upstream/downstream regulators and through the gene transcription mechanism of “classical A/AR signaling” or “non-classical AR signaling”, involving a large number of regulatory factors and signaling pathways. ARs take part in the processes of cancer cell proliferation, migration/invasion, cancer cell stemness, and chemotherapeutic drug resistance. These findings suggest that the AR and related regulators could target the treatment of female reproductive system cancer.

## 1. Introduction

Cancers of the female reproductive system mainly include ovarian cancer (OVCA), endometrial cancer (EC), and cervical cancer (CC). The 2020 Global Cancer Report shows that these three cancers are the most common malignant tumors in women worldwide [1]. Tumors originating from the reproductive system are dependent on steroid hormones. In addition to maintaining the normal development of the female reproductive system as a precursor for the synthesis of estrogen, accumulating evidence consistently supports that the level of androgen (A) and expression of androgen receptor (AR) were abnormal in the female reproductive system in recent years. Reproductive physiology is closely related to pathological phenomena, and A levels are abnormal in OVCA [2,3], EC [4,5] and CC [6,7], suggesting that A is closely related to the occurrence and development of these three gynecological tumors.

The AR is a member of the nuclear receptor superfamily and is a well-defined ligand for A. Target gene expression is activated between A and AR through the gene transcription mechanism of “classical A/AR signaling” [8] or “non-classical A/AR signaling” [9]. Classical AR signaling involves a series of events: (1) AR binding to A and dissociation from heat shock proteins; (2) formation of AR homodimers and translocation to the nucleus; (3) AR dimerization binding to androgen response elements (AREs) within the promoter regions of AR target genes; (4) recruiting coactivators/corepressors; and (5) mediating the transcription of target genes. The non-canonical A/AR signaling pathway is that AR is rapidly activated in the absence of ligands, involving a large number of regulatory factors and signaling pathways. This study reports the association between AR expression and female reproductive system cancers, in which AR is generally abnormally expressed, activates/represses target genes by interacting with multiple factors, and is involved in the regulation of cancer cell proliferation, migration, and drug resistance. The above findings suggest that AR plays a key role in the progression of female reproductive system cancers and may represent a potential therapeutic target for the treatment of these three cancers. Therefore, this review aims to discuss the role of AR and related regulators in the progression of female reproductive system cancer, and to explore clinical therapeutic targets.

## 2. Molecular Biological Characteristics of A and AR

### 2.1. The Type and Origin of A

In females, A is mainly produced by the ovaries and adrenal gland. Biologically active A in the body include testosterone (T) and dihydrotestosterone (5α-dihydrotestosterone, DHT). T can convert to DHT by 5α-reductase. DHT is the most biologically active form of endogenous androgen, and its binding affinity to AR is 2–5 times higher than that of T. A is closely related to female reproductive physiology and pathology, and the main function of A is realized through combination with AR. Furthermore, the level of other androgens links to gynecological cancers. The previous clinical studies found the DHEA, androsterone, and etiocholanolone concentrations were lower in the OVCA patients compared with the age-matched group [10]. Another study, which compared the steroid concentrations in the serum of patients with OVCA, found the androstenedione were significantly higher [11]. Moreover, the free testosterone concentrations were associated with the risk of OVCA specifically in postmenopausal women, while the levels of DHEA and androstenedione were differences in premenopausal women with OVCA [12,13].

### 2.2. The AR Gene Structure and Function of AR

#### 2.2.1. The Gene Structure of AR

AR has 87 kDA-N terminals truncations (absence of the first 187 amino acids), produced by in vitro proteolysis. AR gene is located on X chromosome (loci: Xq11-q12) and consists of eight exons encoding a 110 kDa protein composed of 919 amino acids (Figure 1). AR consists of four functional domains: (1) the N-terminal domain (NTD), encoded by exon 1. There is a highly variable region in NTD, which has transcriptional activity; (2) DNA binding domain (DBD), encoded by exons 2 and 3, is rich in cysteine and zinc fingers, and its main function is to bind DNA. It is the most conservative area; (3) C-terminal ligand binding domain (LBD), encoded by exons 4. LBD is the most frequently studied AR domain, because it combines with ligand to form dimer, which is critical to the activation of AR; (4) the “hinge region” is a small portion of molecule between DBD and LBD. It contains NLS motif (629) RKLKKL (634), which may interact with transcription factors to trigger the receptor to move into the nucleus. In addition, it is a target site for acetylation, ubiquitylation, and methylation [9,14,15]. The constitutively active activation function 1/5 (AF1/5) and ligand-binding dependent activation function 2 (AF2) can bind specific co-regulatory proteins. They are intramolecular head to tail and are crucial for the activity of the full-length receptor [16]. Activation domain 1 (AF-1) is required for complete ligand-activated transcriptional activity. Active domain 5 (AF-5) is responsible for constitutive activity (activity without binding ligands). Based on revealing the AR structure, new competitive binding inhibitors are developed for the treatment of hormone-dependent cancers. It should be noted that AR exon 1 contains two polymorphic trinucleotide repeats: 9–39 CAG repeats (polyglutamine, poly Q) and 14–27 GGN repeats (poly glycine, poly G). Plenty of attention has been paid to AR gene polymorphisms, especially poly Q and poly G, which are related to prostate cancer, breast cancer, lung cancer, etc. [17]. However, the relationship between them and the risk of female reproductive system cancer is still unclear.

#### 2.2.2. Biochemical Mechanism and Functions of A/AR Signaling

##### Classical A/AR Signaling

Unliganded bound AR binds to chaperone complex including Hsp90 and is presented in the cytoplasm [18]. After the DHT or T (preferably DHT) binds to the LBD region of AR, AR changes its conformation, dissociates with the heat-shock protein (HSP), then dimerizes and transfers into nucleus. The activated AR acts as a transcription factor and can directly activate or inhibit A reactivity target genes [9,19,20]. This process is called classical A/AR signaling pathway. The combination of AR and ligand determines the stability of the combination of AR and DNA. The stability affects the amount of initiated transcription. AR interacts with two distinct groups of AREs. cAREs (classical steroid hormone response elements) are the trinucleotides of 5′-TGTTCT-3′ of the AR binding site, partially palindromic repeats. sAREs (selective AREs) are direct repeats of trinucleotide binding elements. After AR binds to AREs, co-activators or co-repressors are recruited to regulate gene expression [21]. AR transcriptional activity is regulated by co-regulatory complexes, such as co-activators and co-suppressors, that bind directly to AR or bind to chromatin via AR to regulate gene activity.

AR is involved in a variety of functional regulations, such as cell proliferation, differentiation, apoptosis, metabolism, and secretion. It is a key factor in maintaining tissue homeostasis. Meanwhile, A plays an important role in the development of various hormone-dependent cancers [17,22], such as prostate cancer [23], EC [24], OVCA [25], CC [26], and so on.

##### Non-Classical AR Signaling (Ligand Independent Pathway)

The non-classical AR signaling pathway is not involved in the regulation of A-responsive gene transcription and translation. The non-canonical AR signaling pathway involves multiple pathways. It mainly includes the regulation of signal transduction pathways, such as MAPK/ERK (Mitogen-activated protein kinase/extracellular regulated protein kinases) activation, and mTOR (Mammalian target of rapamycin) pathway activation through PI3K/Akt (Phosphatidylinositide 3-kinases/Serine/threonine kinases of the AGC family) pathway [27,28]. For example, Src inhibitor can block IL-8-induced AR transcription, indicating that IL-8 induces AR transcription through Src in SKOV-3 cell which is a cell model of epithelial ovarian cancer (EOC) [29]. Moreover, AR is activated in the absence of ligand, and this ligand-independent pathway may be associated with AR phosphorylation or AR-related signaling factors [30]. These pathways cross-regulate each other. It is a fast, transient mode of activation that does not bind to ligands, the response time is only a few seconds to a few minutes (Figure 2).

Src belongs to the tyrosine kinase family and expresses commonly in malignant tumors [31]. Src can induce AR phosphorylation directly and indirectly. Src directly binds to AR to enhance kinase activity, which, in turn, mediates cell cycle progression through the MAPK/ERK/CREB (CREB: cAMP-response element binding protein) signaling pathway. Src interacts with AR, then activates Akt indirectly, which resulting in AR phosphorylation [32]. In addition, Src inhibits AR interaction with co-repressors and promotes AR binding to target genes [23]. In prostate cancer cells, Src is involved in cell survival, proliferation, angiogenesis, invasion, and migration processes [33,34]. In addition to Src, AR interacts with the regulatory subunit p85α of PI3K. p85α is an important oncogenic protein kinase involved in malignant transformation, progression, and migration of cancer [35]. PI3K binds to AR through the SH2 domain of p85α, activates PI3K to generate phosphatidyldiol-3,4,5-triphosphate (PIP3) and Akt, and phosphorylates downstream targets Bad and FKHR-L1 (fork head in rhabdomyosarcoma-L1) [36]. Activation of the PI3K/Akt pathway is ligand independent.

Moreover, there are several AR bypass pathways. In prostate cancer, glucocorticoid receptor (GR) can activate part of AR signaling pathway, and inhibition of GR may be a potential target for the treatment of female reproductive system cancer [37]. Previous research in prostate cancer found that p53 inhibits AR expression by binding to the AR promoter, thereby interfering with AR signaling. MDM2 (murine double minute 2) reduces p53 protein through ubiquitination, and MDM2 regulates AR signaling through p53. Overexpression of N-Myc in prostate cancer cells reduces the expression of AR and its target genes [38]. p66Shc appears to activate mTOR, Akt, ERK, and Rac1, and regulate the activation of migration-related proteins through ROS (reactive oxygen species) in cancer cells, thus it may be involved in cancer cell development and progression [39]. In recent years, scholars have made great progress in identifying the structure–function relationship of AR, AR’s binding partners, and canonical and non-canonical A/AR signaling. It is worth noting that a phase II study has been conducted to evaluate the efficacy of enzalutamide which is an AR antagonist in women with AR positive in OVCA. Results showed that the patients achieved good safety and efficacy which had a higher rate of patients surviving progression-free for at least 6 months (PFS_6_) [40]. AR antagonists show strong potential value in female reproductive system cancer. Therefore, it could be applied to develop new medicines or methods based on these studies to target AR or important factors in its related signaling pathways to control cancer development. 

### 2.3. Cancer of the Female Reproductive System and AR

Female reproductive system cancer is a hormone-dependent cancer. However, the molecular regulation of AR in each type of cancer remains unclear. According to GLOBOCAN 2020 database for 36 cancers in 185 countries, there were 19.3 million new cancer cases and nearly 10 million cancer deaths globally in 2020 (excluding non-melanoma skin cancers). Female reproductive system cancer deaths/new incidences both accounted for 13% of the global total: OVCA (207252/313959), EC (97370/417367), and CC (341831/604127) [1]. The high risk of cancer in the female reproductive system is a serious threat to women’s health.

These three cancers are found in women’s uterus and ovaries which are hormone-dependent organs. Although they differ in organization, they are adjacent to each other and associated with reproductive function. These cancers are speculated to be related. A report shows that young women with EC had an increasing risk of synchronous OVCA, although the prognosis of women with synchronous endometrial and ovarian cancer was good [41]. Estrogen is crucial in both cancers. A as precursors of estrogen, may be involved in the process of cancer. Likewise, patients with endometriosis have a higher risk of OVCA, and AR protein and mRNA have been detected in diseased tissue [42,43]. AR is a well-defined receptor for A in female reproductive system cancers, and AR has attracted great interest as a potential therapeutic target.

#### 2.3.1. OVCA and AR

##### Associations of AR Expressions with OVCA Risks

OVCA is a malignant tumor with the highest mortality rate in female. In 2020, the mortality rate of OVCA patients reached 66% (207252/313959), accounting for 4% of the global cancer deaths [1]. OVCA is a great threat to women’s health and life. OVCA is classified by histopathology into several subtypes. Among them, serous tumors, mucinous tumors, endometrioid tumors, and clear cell tumors are caused by lesions of ovarian surface epithelial cells, collectively referred to as EOC. The incidence rate of EOC was more than 85% [44]. Immunohistochemistry studies have shown that AR expression was detected in EOC approximately 43.5–86% [45,46,47], suggesting that AR may be the screening indicators for EOC. 

The ovary, which produces A, is an important sex hormone synthesis and secretion organ. Recently, the relationship between the A/AR and the OVCA has been explored. Previous studies found that OVCA tissues and cells showed approximately 69~90% AR positive [3,25,48]. AR mRNA upregulated in the EOC cells which were isolated from ascites fluid of advanced primary OVCA patients. The rate of AR positive decreased after chemotherapy [49]. These studies all showed that AR was related to OVCA. By contrast, Hill et al. (2019) study showed that the expression and activity of AR was independent with progression-free survival (PFS) and A in OVCA cell lines [50]. Laura et al. (2022) findings suggested that hormonal status (e.g., pre- and post-menopausal) could influence tumor behavior in high-grade serous ovarian cancer (HGSC) [51]. So, the activity of A/AR may be related to the hormonal status and subtype of patients with OVCA.

##### Associations of AR Gene Polymorphism with OVCA

Interestingly, a large number of studies have found that there was a CAG trinucleotide polymorphism which is about 8–31 repeated segments in exon one of the AR. Genotyping of OVCA cells from 215 Polish–Caucasian patients [52], 1,800 Chinese patients [53], and 594 African American patients [54] showed that the more AR CAG repeats, the less risk of EOC in women. The risk of OVCA was associated with fewer AR CAG repeats and increased AR activity [55]. However, analysis of Ashkenazi Jewish patients [56] and Italian patients [57] found no association between AR repeat length and OVCA. Therefore, the association between the repeated polymorphism and the risk of OVCA varies among ethnicities [58].

##### Molecular Regulation of AR in OVCA

More and more research has focused on the molecular regulation relationship between AR and OVCA. These studies focus on how AR regulates cancer cell proliferation and migration by interacting with a variety of factors in OVCA. These co-regulators including coactivators and corepressors modulate AR activity and activate ARE-mediated transcription in ovarian cancer cells, although most of them also interact with other receptors [59]. OVCA is more common in perimenopause and post-menopause, when the balance of steroids in the ovary favors A. Paradoxically, the expression ratio of AR in ovarian epithelial cells does not change after menopause [60]. It is speculated that the A may act through non-receptor binding pathways. 

AR signaling stimulates proliferation, migration, and invasion of OVCA cells [55]. AR is a key factor in several signaling pathways. It promotes cell proliferation mainly through interactions with many key elements, such as cyclins, transforming growth factor-β (TGF-β) pathway, interleukin-6 (IL-6), epidermal growth factor receptor (EGFR), and various AR co-activators [22,61,62,63]. These findings reveal the important role of A/AR signaling in stimulating the growth and progression of OVCA. A-dependent AR coactivator expression in OVCA may affect other hormonal responses and contribute to OVCA development. For example, A/AR signaling may promote OVCA by reducing TGF-β receptor levels, and allowing OVCA cells to escape the growth-inhibitory effects of TGF-β [64,65,66]. An association exists between EGFR and AR levels in OVCA [62]. EGFR is overexpressed in 30–98% of EOC, and the activation of the EGFR signaling cascade is associated with cell proliferation, migration, invasion, angiogenesis, and resistance to apoptosis [67]. AR has been reported to stimulate the synthesis of EGFR through autocrine or paracrine mechanisms [68]. Western blot analysis of 60 serous cystadenocarcinoma has shown that there was an association between EGFR and AR levels in OVCA [62]. At present, it is still unclear whether there is a signal crossover between the EGFR and AR pathways to accelerate ovarian tumorigenesis. Notably, the brother of the regulator of imprinted sites (BORIS) was reactivated in OVCA and involved in cell proliferation and apoptosis. In OVCA, on the one hand, a reduction in BORIS is associated with a reduction in cell proliferation and viability [69,70]. On the other hand, BORIS a transcriptional repressor of AR binds to the AR promoter to regulate the transcriptional process. Molecular studies have shown that the genes of BORIS, such as CD97, FN1, and FAM129A involved in poor prognosis, chemoresistance and metastasis in OVCA, were associated with AR [71,72]. It suggests that BORIS affect OVCA process by regulating proliferation and apoptosis of cancer cells through AR [73]. Similar studies found that FKBP5 formed a protein complex with the AR, which regulated the transcriptional activity of the two downstream proteins. In OVCA, a 44 kDa AR-interacting protein p44 is a steroid receptor coactivator. It enhances AR-mediated transcriptional activity in a ligand-dependent manner. Moreover, in the presence of A, overexpression of nuclear-localized p44 stimulates proliferation and invasion of OVCA cells [61]. In addition, 5α reductase type I distribute relatively broad and correlate with status of AR. It suggests that the Isozyme plays an important role in metabolism of A in the human EOC [74]. It has been reported that small glutamine-rich tetratricopeptide repeat-containing protein alpha (SGTA) affected AR signaling in OVCA cells. The signaling may decrease with the development of serous ovarian cancer, without the progression of metastatic [75]. Transcriptome sequencing results showed that circ_100395 promoted the expression of p53 by regulating miR-1228, which might inhibit the growth and metastasis of OVCA through the non-classical-AR pathway [76]. It is worth noting that methyltrienolone which is an AR ligand can increase the ability of cell invasion in AR-positive OVCA lines. These findings indicated that AR promoted cell proliferation/invasion of OVCA.

##### AR Signaling in OVCA Stemness

Generally, cancer stem/progenitor cells (CSPCs) are considered to be responsible for cancer phenotypes, including stemness/pluripotency, metastasis, drug resistance, and high recurrence rates to stimulate tumor growth and disease progression [77]. Numerous studies have shown that AR regulated the progression of cancer stem cells in various cancer types, including OVCA. It has been demonstrated in ovarian teratocarcinoma (OVTC) cells that AR utilized a ligand-independent pathway to promote OVTC cell growth in CSPCs, such as CD133 cells [40]. The interaction of Nanog (a stemness marker gene) with the AR signaling axis can activate Nanog promoter transcription, which may induce or contribute to ovarian cancer stem cells (OCSCs) proliferation, migration, spheroid formation and colony formation [78]. These studies suggest that AR expression promotes OVCA proliferation and migration through canonical and non-canonical signaling pathways.

##### Clinical Trials of Targeting A/AR Therapy in OVCA

Association of AR and Chemoresistance of OVCA

Currently, chemoresistance is a major challenge to cancer chemotherapy. The first-line therapy for OVCA is surgery, and the second-line therapy is chemotherapy (primarily platinum or paclitaxel (Taxol)). In the process of chemotherapy, OVCA patients are prone to develop drug resistance, and the overall survival of patients is shortened. Therefore, paclitaxel resistance (Taxol resistance, txr) has raised great interest. Analysis of genes involved in txr revealed that ERAR is a key gene for resistance [79]. AR expression promotes the development of cisplatin resistance in EC cells [80]. FK506-binding protein 5 (FKBP5), a member of the immunoaffinity protein family, has peptidylalanyl cis/trans isomerase (PPIase) activity and acts as a scaffold protein, which recruits AR and regulates gene expression. The Akt kinase pathway is regulated by FKBP5, and the FKBP5/AR complex may affect the sensitivity of cancer cells to paclitaxel by regulating the expression of the txr gene [81]. Dimethylcurcumin (ASC-J9) is an anti-AR drug that selectively degrades and inhibits AR transcriptional activity by disrupting the interaction between AR and its regulators [82]. In OVCA, paclitaxel can bind to the proximal promoter region of ABCG2, an MDR-related membrane protein, and enhance AR transcriptional activity, resulting in increased txr in EOC serous subtype cell lines. Targeting the AR with an AR degradation enhancer (ASC-J9) resensitized txr of EOC in vitro and in vivo, reducing drug resistance [83,84]. Molecular studies have shown that the Toll-like receptor 4 (TLR4) gene was associated with the AR gene in OVCA cells (SKOV3). DCDC2, ANKRD18B, ALDH1A1, c14orf105, ITGBL1, and NEB were related to AR. These six genes were identified as TLR4 and AR-regulated genes involved in txr [85]. Subsequent studies were performed on overexpressing of TLR4 in OVCA cells treated with paclitaxel. Since IL-6 is a central gene among upregulated genes, nuclear translocation of AR is induced by IL-6, which could promote the growth of EOC by activating the AR gene promoter [24]. It was found that knockdown of endogenous IL-6 reduced AR and TLR4 expression in txr OVCA cells. It is indicated that TLR4 and IL-6 play crucial roles in AR gene regulation and function of txr in OVCA cells. Interferon regulatory factor Ⅰ (IRF Ⅰ) was also found as a downstream target of IL-6 signaling and a regulator of AR expression [63]. These results suggest that due to the reducing of paclitaxel resistance in OVCA cells by AR, targeting degradation of AR protein may be more beneficial than antiandrogens for the treatment of OVCA.

Association of OVCA Cell Migration and Prognosis

Numerous studies have demonstrated that high migration rates of cancer cells were often associated with poor prognosis. Clinical studies have shown that matrix metalloproteinase-2 (MMP-2) was associated with AR and reduced overall survival in 88 patients with EOC in Mexico. It was speculated that in the presence of AR, MMP-2 in EOC cells involved in basal layer degradation and promoted ovarian cell adhesion to the peritoneum and momentum, promoting tumor cell migration [86]. In a study, when the hedgehog signaling pathway was inappropriately activated in OVCA, the two pathways end effector GLI3 and AR interacted physically and dependent functionally on each other to promote the growth and migration of malignant tumor cells. The GLI3 expression was negatively associated with overall survival in OVCA patients [87]. Methylation of melanoma associated antigen 11 (MAGEA11) which was a co-activator of AR was related to poor prognosis of cancer [88]. AR overexpression promoted the proliferation and migration of OVCA cell lines, resulting in a more aggressive OVCA phenotype [89]. An evaluation of the survival of 118 serous and endometrioid OVCA patients showed that tumors co-expressing PR and AR were most favorable for patient prognosis [90]. Surprisingly, the incidence of brain metastases from EOC increases, and there was a correlation between AR loss and central nervous system (CNS) localization [91]. Low expression of AR was associated with a higher risk of cancer brain metastases. When AR expression was less than 10%, the risk of developing brain metastases increased by more than 9 times [92], indicating that AR was related to cell migration performance and affected the prognosis and overall survival of patients.

Recently, there are encouraging clinical results on AR antagonist. A Phase II study has been conducted to evaluate the efficacy of enzalutamide in women with AR positive ovarian cancer [93]. Enzalutamide is an oral, small-molecule, second-generation AR antagonist that works through three mechanisms: blocking androgen binds to AR, blocking AR nuclear translocation, and inhibiting AR bind with nuclear DNA and coactivator recruitment [94]. It was approved by the U.S. Food and Drug Administration in 2012 to treat castration-resistant prostate cancer. The study endpoints were to estimate the proportion of patients surviving progression-free for at least 6 months (PFS_6_). The patients experienced multiple recurrences with recurrent AR-positive (AR+) ovarian cancer who received daily enzalutamide 160 mg until progression of disease or treatment discontinuation. Assessing long-term survivors of epithelial ovarian cancer, the PFS_6_ was observed in at least 13 of 59 patients (22%). Including 38.5% of patients with LGS met the PFS_6_ endpoint. In addition, the patients showed toleration with enzalutamide, while a phase II study about the efficacy of anti-androgen and gonadotropin-releasing hormone with bicalutamide and goserelin in patients suffering from epithelial ovarian cancer found no survival benefit [95]. It should be noted that the enrolled patients, of whom only 58% were AR+ without been selected, received bicalutamide which was a less potent, first-generation AR antagonist [94]. Therefore, selected high AR+ patients with ovarian cancer and the more potent AR antagonist enzalutamide achieved good safety and efficacy. Therefore, AR targeting has the potential to treat OVCA.

However, so far, clinical trials on antiandrogen receptor for OVCA have been relatively non-randomized and undersized to draw reliable conclusions. Further clinical trials with larger sample sizes and randomized designs are needed.

#### 2.3.2. EC and AR

##### Associations of AR Expressions with EC Risks

EC is one of the most common malignancies in women, with low mortality due to its obvious early symptoms [96]. EC originates from the endometrium, which is a multicellular tissue. Endometrial cells are sensitive to sex hormones which regulate the process of cell proliferation, differentiation, and apoptosis. Dysregulation of these processes result in endometrial lesions [97]. EC is graded according to the FIGO system and is generally divided into two subtypes: estrogen-dependent type I and estrogen-independent type II (less common clinically, but more aggressive). About 95% of endometrial tumors are adenocarcinomas due to malignant transformation of endometrial glandular epithelium. The incidence of EC varies across regions [98] and age. EC is more common in perimenopausal and postmenopausal women (between 50 and 65 years of age) [99]. During this period, estrogen is mainly converted from A. As estrogen plays an important role in the occurrence and progression of EC, the roles of A and AR on EC have attracted much attention.

In 1978, AR was found in EC, and 65–80% of EC was positive for AR [4], while the other study revealed that 20.2% of EC cases showed positive AR expression [100]. The expression of AR may be related to histological subtypes or staging of cancer. In the serum of post-menopausal women, circulating levels of androstenedione and free testosterone were associated with the risk of EC [101,102,103]. Genetic variation analysis showed that AR was significantly associated with an increased risk of EC [104]. However, little is known about the role of AR in EC.

##### Associations of AR Gene Polymorphism with EC

Similar to OVCA, the polymorphism of CAG repeats in AR exon 1 is of interest. Some studies suggested that CAG repeat length had nothing to do with EC [105], while others showed that CAG repeat length was negatively correlated with AR activation ability [106]. AR activity decreased by 1.7% for each additional glutamine repeat [107]. Because CAG repeat affected the ability of AR to recruit co-activators and other transcription factors, which, in turn, affected AR stability and activity [108]. Results from genotyping of tissue from 204 EC patients supported this argument, indicating an association between the number of repeats of CAG and GGN in the NTD of AR and the progression of EC. There are CAG shorter repeats of AR in benign tumors [108,109]. Such contradictory results may be related to ethnic differences in the selected cases. In addition, the increase repeats of GGC in AR may be related to EC [110].

Hypermethylation of the CpG region of the AR gene spanning the transcription start site is associated with stage III and IV endometrial malignancies [111]. Presumably, hypermethylation of the CpG island of the AR gene leads to AR inactivation in EC [111]. Mismatch repair (MMR) deficiency is one of the most common molecular alterations in EC, caused by methylation of the MLH1 gene. Analysis of MMR-deficient tissue in grade two EC found that MMR deficiency was significantly associated with reduced AR expression. It is speculated that the reduced expression of AR may lead to the methylation of MLH1 [112], which, in turn, causes MMR, leading to EC.

##### Molecular Regulation of AR in EC

AR-dependent signaling inhibits the proliferation of endometrial tumor cells [113]. AR can inhibit the growth of endometrial epithelial cells in vitro, and the inhibition is blocked by AR antagonists [114]. AR may promote EC by increasing CD133 expression, cell migration and epithelial–mesenchymal transition [80]. As epithelial cancers, EC is most closely associated with genomic alterations in the PI3K pathway of all cancers. In PI3K pathway genes and/or gene products, 70% of EC patients, and more than 50% of CC patients have at least one mutation, of which PTEN was lost in 49% of EC, and PI3Ks catalytic subunit (PIK3CA) was mutated in 37% of endometrial and 29% of CC. Genetic analysis showed that mutations in PIK3CA, PTEN, and Akt1 were more frequent when AR was overexpressed, and AR was overexpressed in 29% of PIK3CA-mutated cases. It indicated that there is a strong association between the PI3K/Akt/mTOR signaling pathway and the AR signaling axis [115]. AR regulates the expression of cyclin D1 (CCND1) which is overexpressed in EC [116]. The expression decreased with EC progression, suggesting that AR-regulated prostate-specific membrane antigen (PSMA) decline may play an important role in EC disease progression [117]. Comparing the tissue expression patterns of 85 postmenopausal EC patients with healthy hyperplastic endometrium patients, immunohistochemistry and qPCR methods, results showed that the retention of progesterone receptors/ ERα/Erβ (estrogen receptor subtype) and the loss of AR might promote the uncontrolled growth of high-grade endometrial carcinoma (HGEC) [118], suggested that AR deletion was important for HGEC cell proliferation.

The number of AR appears to correlate with the degree of tumor differentiation [119]. By immunohistochemical methods, it was verified that AR is expressed in HGEC [120]. AR expression was associated with type I, early tumor stage (I–II), and low FIGO grade (1–2) of EC, and is significantly associated with lymphatic vascular invasion (*p* = 0.041) [5]. Enhancement of epidermal growth factor (EGF) has been found to upregulated genes involved in the IGF-1 and Wnt signaling pathways, resulting in that A promoting cancer cell proliferation [121,122]. Paradoxically, some studies found that A inhibited endometrial cell proliferation in vitro [123]. MFE-296 is the AR weakly positive EC cell line where A inhibited cell growth [124]. This means that AR played different roles in different EC subtypes. In addition, the ratio of AR to ERα related to survival rates of patients. The survival rate of patients with high AR/ERα was poor, indicating that ERα status might affect the effect of AR. AR is more expressed in metastatic EC [125].

It has been reported that A and AR might be involved in the process of endometrial cell proliferation and inhibition of apoptosis by regulating the expression of insulin growth factor I (IGF-I) which was an A-responsive target gene in utero [126,127]. The IGF system was thought to regulate steroid hormone action in the endometrium through autocrine and paracrine mechanisms. Unantagonized IGF-I action might increase EC risk as it favors endometrial cell proliferation and inhibits apoptosis [128]. Combination therapy with the hypoglycemic drug metformin and the contraceptive Diane-35 (a progesterone agonist, an inhibitor) had the potential to restore EC in early-stage EC polycystic ovary syndrome-insulin resistance (PCOS-IR) patients’ normal endometrial cells. However, the specific molecular mechanism is still unclear. It is speculated that metformin treatment reduces cell proliferation and increases apoptosis in uterine serous carcinoma by inhibiting the insulin/IGF-I signaling pathway. Diane-35 (containing two components, an estrogen drug and an anti-androgen drug) increases progesterone action and enhances PR-mediated IGFBP-1 activity while inhibits AR signaling in the endometrium [129]. Treatment with the progesterone receptor antagonist RU 486 increased epithelial and stromal AR expression in human endometrial tissue [130], suggesting that the inhibitory effect of progesterone on endometrial AR expression was mediated by PR.

The proto-oncogenic transcription factor c-Myc has been found to promote tumor growth in various AR-related cancers through AR signaling [131,132]. Aberrant activation of c-Myc in EC was associated with EC cell proliferation [133,134]. The histone demethylase KDM4 regulated AR transcriptional activity and the depletion of KDM4 protein inhibited AR-mediated transcription [135]. Members of the KDM4 family, KDM4B and KDM4A, are master regulators of AR transcriptional activity in ECs. KDM4B as a key co-activator of c-Myc directly enhances c-Myc-mediated metabolism [136]. Molecular studies have shown that overexpression of KDM4B promoted AR recruitment to the c-Myc (MYC) enhancer and induced H3K9 demethylation, increasing AR-dependent transcription of c-Myc mRNA [137]. However, KDM4A promoted progression of EC by regulating AR activity in different EC subtypes. In MFE-296 cell line, KDM4B and AR up-regulated c-Myc expression. KDM4B was positively correlated with AR high expression, while in AN3CA cells, KDM4A and AR down-regulated p27kip1, and KDM4A were positively correlated with AR low expression. The findings suggest that KDM4B and KDM4A promote EC progression by regulating AR activity [138]. According to Bai et al. (2008), the melanoma antigen gene protein 11 (MAGE-11) was an AR regulator. It increased AR transcriptional activity by selectively binding to the FXXLF motif and regulating N/C binding, exposing AF2 in the ligand-binding domain to coactivators. Epidermal growth factor (EGF) signaling enhanced the transcriptional activity of A-dependent AR by post-translational modification via MAGE-11 [139]. In addition, curcumin inhibited the proliferation and apoptosis of human EC cells by downregulating their AR expression through the Wnt signal pathway [140].

##### Clinical Trials of Targeting A/AR Therapy in EC

Indole-3-methanol (I3C) inhibits the activation of multiple transcription factors including AR, the synthesis of DNA carcinogens and free radical production. It induces cell apoptosis, stimulates 2-hydroxylation of estradiol, and inhibits cancer cell invasion and angiogenesis. Therefore, I3C can inhibit the proliferation of EC, breast cancer, and other tumor cells in vitro [141].

Estrogen induced AR expression [127]. An anastrozole (aromatase inhibitor) arm study showed that the significant decrease in the glandular expression of ERα and AR could restrict the proliferation of cancer cell. Enzalutamide as an AR inhibitor can inhibit cancer cell growth in uterine leiomyosarcoma [142]. mTOR promotes AR receptor phosphorylation and activates AR transcription in a ligand-independent manner [143], and inhibition of mTOR reduces protein translation and prevents abnormal cell proliferation and tumor angiogenesis. Sapanisertib is a highly selective mTOR kinase inhibitor that inhibits both mTORC1 and mTORC2. Preliminary antitumor activity of sapanisertib was observed in EC [144]. It is suggested that A and AR inhibitors could be used to treat EC. Emerging data have demonstrated promising efficacy of enzalutamide in combination with chemotherapy. The phase II ENPAC trial investigated ORR and PFS_6_ in 35 patients with endometrial cancer. Findings demonstrated the safety and promising efficacy of enzalutamide in this setting, with a 71% ORR (95% CI: 54–85%) and 83% PFS_6_ rate (95% CI: 66–92%).

##### Association between AR and Stemness and Chemoresistance of EC 

The evaluation of the therapeutic effected of tumor cells in vitro showed that increased AR expression promoted the migration and epithelial–mesenchymal transition process of EC cells and CSPCs. It also reduced the cytotoxic effect of cisplatin on EC cells [80].

Further research on the occurrence and development of AR and its regulatory factors will help to deepen the understanding of EC, discover new biomarkers and drug targets, and provide new strategies for cancer treatment.

#### 2.3.3. CC and AR

##### Associations of AR Expressions with CC Risks

CC is the fourth most frequently diagnosed cancer and the fourth leading cause of cancer death in women. According to the 2020 global female cancer statistics, the incidence of CC was 6.5% and the mortality rate was 7.7% [1]. The two main types of CC are squamous cell carcinoma (SCC) and adenocarcinoma. More than 90% of patients suffer SCC [145]. The main cause of CC is persistent infection with human papilloma virus (HPV), and the HPV types are mainly HPV16 and HPV18 [146]. Over the past decade, the global incidence of CC had declined dramatically with the help of two methods of CC prevention: regular cervical screening and HPV vaccination [147]. At present, the main methods for the treatment of CC are debridement surgery, post-operative radiotherapy and chemotherapy. However, due to cancer cell metastasis and high lymphatic metastasis, the poor prognosis and low survival rate of CC remains a problem [148]. 

AR-dependent signaling affected uterine growth and ovarian function, and AR was positive varies in different CC subtypes [6]. Immunohistochemistry studies have shown that 100% of AR was expressed in normal epithelium and low grade cervical intraepithelial neoplasia (LSIL-CIN1) (*n* = 30). However, AR expression was observed only in 63% of high-grade cervical intraepithelial neoplasia (HSIL-C 2/3) (*n* = 30) and 23% of infiltrating squamous cell carcinoma (ISCC) (*n* = 13). Loss of AR expression was common in HSIL and ISCC. It is prevalent in high-grade squamous intraepithelial lesions and ISCC due to the complex interaction between high-risk HPV and AR [7]. However, existing studies on CC and steroid hormones mainly focus on estrogen/progesterone and their receptors, there are few reports on the association between AR and CC.

##### Molecular Regulation of AR in CC

A next-generation sequencing (NGS) analysis of the cancer genome maps in 182 CC patients was conducted and results showed that AR amplifications, mutations, and deletions in 7% of patients [149], indicating that AR has an unknown significance in CC.

The ASXL family are epigenetic scaffold proteins that assemble epigenetic regulators and transcription factors into specific genomic loci with histone modifications. They are thought to be tumor suppressor or oncogenic, and play different roles in different types of tumors. ASXL1 assembles various proteins based on protein–protein interactions. It has been found that ASXL1 interacted directly with AR, and promoted AR-dependent transcriptional activation. In CC, ASXL1 expression was increased [150]. It was speculated that ASXL1 exerts oncogenicity by activating AR transcriptional activity. Human nuclear receptor interacting protein (NRIP) was a ligand-dependent co-activator of AR, and RNA interference sequences targeting NRIP genes (5′-GATGATACAGCACGAGAAC-3′) significantly reduced the proliferation of CC cells (C-33A) [151]. It was speculated that NRIP might promote cancer cell proliferation by enhancing the transcriptional activity of AR. miRNAs are a class of short, highly conserved noncoding RNAs that regulate gene expression by repressing translation or inducing mRNA degradation post-transcriptionally [152]. Abnormal expression of miRNAs has been reported in various tumors [153], for example EC [154], OVCA [154], and CC [155]. Erα and AR were direct target of miR-130a-3p, and studies have found that loss of Erα/AR played a major role in CC progression. The microRNA (Mir-130A-3p) expression level in CC tissues was higher, and Mir-130a could directly bind to the 3’UTR of ESR1 and AR mRNA, suggesting that miR-130a functionally regulated Erα/AR-mediated CC progression and promoted CC cell proliferation and invasion. Interfering with miR-130a-3p, overexpression of Erα/AR significantly inhibited CC cell proliferation and invasion, and antagomiR-130a reduced CC tumor size and weight in vivo. Therefore, the results suggested that miR-130a-3p might promote CC progression by inhibiting ERα and AR [156]. The expression of AR protein in the vaginal wall of surviving female patients who received CC radiotherapy was decreased [157], suggesting that decreased hormone receptor expression may influence CC progression. 17APAD is an androstane derivative with a bulky subunit that prevents A from binding to AR. After exposure of two CC cell lines (SiHa and C-33A cells) to 17APAD16 for 48 h, CC cells exhibited proliferative, migration-inhibitory, and invasive properties. It indicated that the combination of A and AR were very important for the proliferation, migration, and invasion of CC cells [158]. Cyclin pathway genomic abnormalities are common in human solid tumors, with AR and cyclin activation/sensitization alterations occurring more frequently in CC (compared to AR alterations and wild-type cyclin activation/sensitization alterations), suggesting a possible association between AR and cyclins in CC. However, there was no relationship between the prognosis of CC patients and the presence of AR [159,160]. Therefore, research is needed to further our understanding of the relationship between AR and CC.

## 3. Conclusions

Cancers of the female reproductive system are a serious threat to women’s health due to their high morbidity and mortality. AR plays an important role on steroid-dependent cancers, especially in pre- and post-menopausal. AR-based combination therapies for the treatment of female reproductive system cancers have gradually become a hot research topic. For example, AR antagonists, such as enzalutamide and bicalutamide, have been used in clinical trials. Enzalutamide has good tolerability and appropriate safety in patients with AR+ recurrent ovarian cancer, especially in the AR+ LGS subpopulation. So, AR antagonists may be a potential treatment option for the specific subtypes of female reproductive system cancers. Although efforts have been made to explore the effects and mechanisms of AR on female reproductive system cancer, further research needs to be conducted in preclinical trials.

This review provides a comprehensive overview of AR expression and polymorphisms, AR coregulators, and related mechanisms in the female reproductive system cancers. Findings from clinical, pharmacological, and genetic studies have now converged to demonstrate an important role of AR in these cancers. The mechanisms of AR in these cancers are complex and dynamic, and require the cooperation of many coregulators which are not all identified. In addition, AR is involved in the proliferation and migration of cancer cells and chemotherapeutic drug resistance. Thus, more attention should be paid to AR target genes, activation mechanisms and functions. Further work is required to develop appropriate therapeutic strategies for female reproductive system cancers. Particularly, novel biomarkers and drugs, which degrade or inactivate AR protein through AR co-regulators or antagonists, should be explored. 

## Figures and Tables

**Figure 1 ijms-23-07556-f001:**
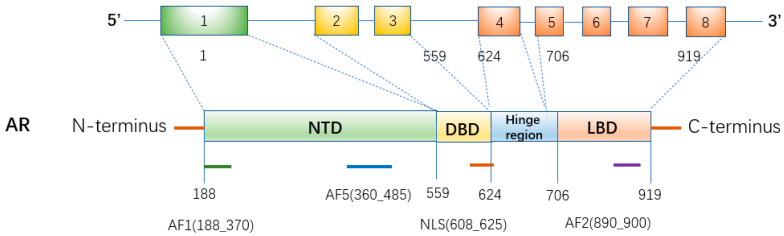
Domain diagram of two isotypes of androgen receptor. AR gene contains eight exons: exon 1 encodes n-terminal domain (NTD); exon 2 and 3 encode DNA binding domain (DBD). Exon 4-8 encodes the hinge region and c-terminal ligand binding domain (LBD). The hinge region containing the nuclear localization signal is encoded by the 5′ region of exon 4. Activation function (AF) is active functional domain. NLS is nuclear localization signal (NLS). The lower numbers refer to amino acid residues that separate domains from the N-terminal (left) to the C-terminal (right).

**Figure 2 ijms-23-07556-f002:**
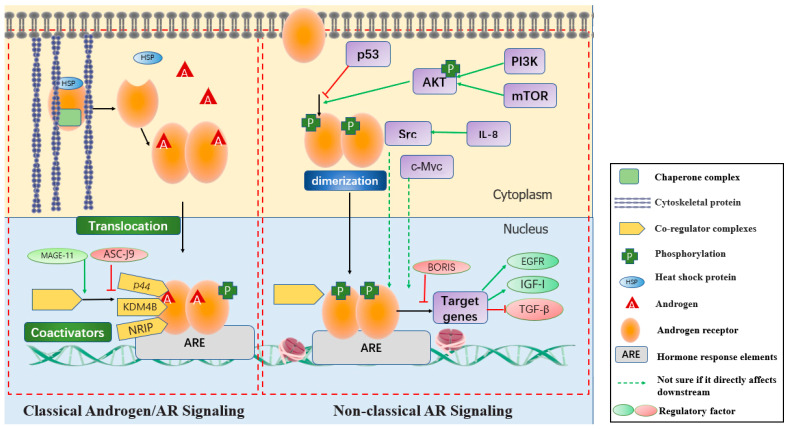
Schematic of classical and non-classical androgen(A)/androgen receptor (AR) signaling. The red box on the left: AR is immobilized in the cytoplasm by actin. AR dissociates from heat shock protein (HSP) and binds to A, forming A dimer and translocation into the nucleus. Co-regulators (KDM4B, NRIP, etc.) regulate transcriptional activity of AR at androgen response elements (ARE) and AR mediates transcription of downstream target genes. MAGE-11: Melanoma antigen gene protein 11; ASC-J9: Dimethylcurcumin; p44: A 44 kDa AR-interacting protein; KDM4B: The histone demethylase 4B; NRIP: Nuclear receptor interacting protein. The red box on the right: AR is activated in a non-ligand-dependent manner under the interaction of multiple regulatory factors (AKT, Src, etc.), mediating transcription of downstream target genes. p53: A tumor suppressor gene; AKT: Serine/threonine kinases of the AGC family; FoxO: Forkhead box O; Src: Tyrosine protein kinase; PI3K: Phosphatidylinositide 3-kinases; mTOR: Mammalian target of rapamycin; IL-8: Interleukin-8; c-Myc: A transcription factor; BORIS: Brother of the regulator of imprinted sites; EGFR: Epidermal growth factor receptor; IGF-I: Insulin growth factor I; TGF-β: transforming growth factor-β.

## Data Availability

Not applicable.

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
