# Peer review of "Molecular Regulation of Androgen Receptors in Major Female Reproductive System Cancers"

_ijms, 2022, doi:10.3390/ijms23147556_

Round 1

Reviewer 1 Report

The title of the review by Wu et al. suggests a summary of the molecular pathways of the androgen receptor in female reproductive cancers.

In addition to molecular pathways known for AR, the cancer specific information about androgens and its receptor is described.

This review has many wrong statements and is very hard to follow authors conclusions.

  1. Authors mention two receptors while one is detected in vitro only. That clearly means an in vitro artifact of an AR-B form.

Thus delete this point including the title. In other extract preparations even additional AR bands appear which are in vitro artifacts. Naturally shorter forms of AR exist as splice variants.

  1. The main molecular pathways of AR signaling described here derive from knowledge obtained from prostate cancer and not from female reproductive cancers. Authors do not differentiate this point and thus mislead the readership.

  1. In abstract only coactivation is indicated. However, RNA-seq and molecular analyses show that also corepressors regulate AR activity and many AR target genes are repressed by androgens. This part is ignored in the introduction.

  1. The AR dimerizes prior nuclear translocation. Authors indicate it wrongly.

  1. Authors did not mention clinical trials that use AR ligands for female reproductive cancers.

Many other sentences are also wrong. I mention only few here. Many other wrong statments are spread in the entire text. It seems nobody critically had read the manuscript.

  1. Lane 70: “ T in the female body synthesizes estrogen…“ This is wrong T cannot synthesize estrogen! Only the aromatase can sythesize estrogen from T.

  1. Wrong statement:

„Src belongs to the non-tyrosine kinase“ Src is a tyrosine kinase!

Src belongs to non-receptor tyrosine kinase“

  1. Lane 66: “Female A is mainly produced by the ovaries.“ Wrong statement! It must state: In females, A is mainly produced by the ovaries“

  1. Lack of critical perspectives:

Based on the very limited knowledge on AR activity in female reproductive cancers, authors mention:

„Currently, there is an urgent need to develop new drugs.. to target AR …..  to control cancer development.“

However authors do not show enough evidence why urgent need to target AR.

Specifically in these contexts:

  • AR expression and activity do not predict the dependence of OVCA cell lines on A growth. AR expression and activity do not correlate with progression-free survival.
  • Low expression of AR is associated with a higher risk of cancer brain metastases.
  • - Only 20.2% of EC cases showed positive AR expression.
  1. Lane 87: „Among DBD and LBD. The….“ Unclear what is meant.
  2. Lane 79: Please write in internationally accepted form: „X chromosome (loci: XQ11-Q12)“ Must be written as: X chromosome (loci: Xq11-q12)

Author Response

We would like to thank you for your careful reading, helpful comments, and constructive suggestions, which has significantly improved the presentation of our manuscript.

We have carefully considered all comments from the reviewers and revised our manuscript accordingly. In the following section, we summarize our responses to each comment from the reviewers. We believe that our responses have well addressed all concerns from the reviewers. We hope our revised manuscript can be accepted for publication.

  1. Authors mention two receptors while one is detected in vitro only. That clearly means an in vitro artifact of an AR-B form.

Response: We have deleted the content involving AR-B form, including the title2.1. and 2.2. See line 78、79 and Figure 1.

  1. The main molecular pathways of AR signaling described here derive from knowledge obtained from prostate cancer and not from female reproductive cancers. Authors do not differentiate this point and thus mislead the readership.

Response: I have distinguished the main molecular pathways of AR signal transduction described in Figer2. Some of which were confirmed to be derived from female reproductive system were retained, while those derived from prostate cancer were deleted. In addition, it is clearly identified in the text that this molecular pathway is derived from knowledge of prostate cancer rather than female reproductive cancer, see line 201, 211, 215.

  1. In abstract only coactivation is indicated. However, RNA-seq and molecular analyses show that also corepressors regulate AR activity and many AR target genes are repressed by androgens. This part is ignored in the introduction.

Response: The content of corepressors was added in the abstract , see line 25.

  1. The AR dimerizes prior nuclear translocation. Authors indicate it wrongly.

Response: We have corrected the location of dimerization to be in the nucleus, see Figure 2. Accordingly, we have corrected the part involved in the article, see line 57.

  1. Authors did not mention clinical trials that use AR ligands for female reproductive cancers.

Response: We have added clinical trials that use AR ligands (e.g. AR antagonist) for female reproductive cancers, see references 86, 87, 89, 91.

  1. Lane 70: “ T in the female body synthesizes estrogen…“

Response: We have corrected the sentence to ” In females, A is mainly produced by the ovaries and adrenal gland.” see line 72.

  1. „Src belongs to the non-tyrosine kinase“ Src is a tyrosine kinase!

Response: We have corrected the sentence to “Src belongs to the tyrosine kinase family”. Sorry for wasting your precious time on the detail. See line 195.

  1. Lane 66: “Female A is mainly produced by the ovaries.“ Wrong statement! It must state: In females, A is mainly produced by the ovaries“

Response: We have corrected the sentence. See line 72.

  1. Lack of critical perspectives:

Based on the very limited knowledge on AR activity in female reproductive cancers, authors mention:Currently, there is an urgent need to develop new drugs.. to target AR …..  to control cancer development.“

However authors do not show enough evidence why urgent need to target AR. Specifically in these contexts:

①AR expression and activity do not predict the dependence of OVCA cell lines on A growth. AR expression and activity do not correlate with progression-free survival.

②Low expression of AR is associated with a higher risk of cancer brain metastases.

③Only 20.2% of EC cases showed positive AR expression.

Response: To comprehensively analyze AR activity in female reproductive system cancers, we refer to multiple viewpoints in this paper.

 First, numerous studies have shown that AR is associated with cancers of the female reproductive system. For those results mentioned on the ①②③, we supplemented the literature on the association of AR with OVCA and EC, showing that hormonal status (e.g., pre- and postmenopausal) could influence tumor behavior in specific cancer subtypes. It is suggested that the activity of A/AR may be related to hormonal status and cancer subtype. See reference 47, 98, 99, 100.

Secondly, involving the urgent need to target AR, we supplemented the reference on clinical trials that use AR antagonist in OVCA. Results showed AR antagonist enzalutamide achieved good safety and efficacy. Targeting AR has the potential to treat OVCA. See reference 91.

  1. Lane 87: „Among DBD and LBD. The….“ Unclear what is meant.

Response: We have deleted the sentence. See line 90.

  1. Lane 79: Please write in internationally accepted form: „X chromosome (loci: XQ11-Q12)“ Must be written as: X chromosome (loci: Xq11-q12)

Response: We have corrected the form. See line 82.

The details of specific revisions are changed with revision mode. Thank you again.

Reviewer 2 Report

This review on AR in different gynaecological cancers provides an important topic of scientific interest, but prior to consideration of publication I believe the work need significant additional edits.

The basic mechanisms described between pages 1-4 are fine. Major concerns however start at page 5:

  • the subsection 2.3 shows an unnecessary depiction of incidence and mortality in gynaecological cancers as well as too basic (even lay level basic) depiction of gynae cancers in figure 3. I suggest to remove both figures.
  • I am concerned in regards to the statement in line 187-188 as it is not correct from a clinical point of view (EC and risk of OVCA) - there is a specific subset of population this might be true, but not as a general statement.
  • Overall for all three cancers a lot of time is used to describe histological subtypes, but little is then done to connect these subtypes to AR  - if you spend such detail on the pathohistology address the pathohistology or focus on the other clinically relevant parts.
  • The analysis in all three cancers usually follows a similar pattern of analysis - consider rather grouping the knowledge on paragraphs of eg expression levels, genetics ... for all three cancer types. This would then make it easier to directly compare the differences and current conclusions on this topic together
  • This leads to the next point, this is that this is a highly complex paper with a lot of data and would therefore need a discussion section of the implications of your analysis.
  • Furthermore, the clarity of the paper would improve immensely if you would structure an analysis of studies (based on subsets, methodologies) for AR.

Author Response

We are grateful to the editors for their careful review of our manuscript, your helpful comments and constructive suggestions have greatly helped us to improve the presentation of the manuscript.

We have carefully considered all the comments of the reviewers and have revised our manuscript accordingly. In the next section, we summarize our responses to each of the reviewers' comments. We hope that our revised manuscript will be accepted for publication.

  1. the subsection 2.3 shows an unnecessary depiction of incidence and mortality in gynaecological cancers as well as too basic (even lay level basic) depiction of gynae cancers in figure 3. I suggest to remove both figures.

Response: We have deleted the figure 3 A, B. See Figure 3.

  1. I am concerned in regards to the statement in line 187-188 as it is not correct from a clinical point of view (EC and risk of OVCA) - there is a specific subset of population this might be true, but not as a general statement.

Response: We have corrected the sentence to “A report shows that the young women with EC have an increased risk of synchronous OVCA, although the prognosis of women with synchronous endometrial and ovarian cancer is good”. It is based on the results of reference 37. See line 272-274 and reference 37.

  1. Overall for all three cancers a lot of time is used to describe histological subtypes, but little is then done to connect these subtypes to AR - if you spend such detail on the pathohistology address the pathohistology or focus on the other clinically relevant parts.

Response: We removed unnecessary descriptions of histological subtypes. And in the subsequent description, according to the description of the reference, we try to correspond the subtype with AR. In addition, we supplemented the reference on clinical trials that use AR antagonist in OVCA. See references 86, 87, 89, 91.

  1. The analysis in all three cancers usually follows a similar pattern of analysis - consider rather grouping the knowledge on paragraphs of eg expression levels, genetics ... for all three cancer types. This would then make it easier to directly compare the differences and current conclusions on this topic together

Response: For the convenience of readers, we have divided the paragraphs into the following groups: associations of AR expressions with OVCA/EC/CC risks, associations of AR gene polymorphism with OVCA/EC, molecular regulation of AR in OVCA/EC/CC, clinical trials of targeting A/AR therapy in OVCA/EC.

  1. This leads to the next point, this is that this is a highly complex paper with a lot of data and would therefore need a discussion section of the implications of your analysis.

Response: We added a discussion at the end of each group. Besides, the conclusion section of the article has been rewritten.

  1. Furthermore, the clarity of the paper would improve immensely if you would structure an analysis of studies (based on subsets, methodologies) for AR.

Response: For the convenience of readers, on the basis of grouping in the previous step, the content is classified. For example, OVCA is subdivided into: AR signaling in OVCA stemness, association of OVCA cell with migration and prognosis, association of AR with chemoresistance of OVCA.

The details of specific revisions are changed with revision mode. Thank you again.

Round 2

Reviewer 1 Report

The revised version has tremedously improved. Some minor points should be addressed prior publication:

Page 2, line 53 and page 4, line 120:  The order should be that the AR dimerizes first then translocates. The text lists the steps in a different order. Please change the order.

Line 664: Please correct these words:

„enzlutrimide“ must be corrected to „enzalutamide“

What is meant with „kalutrimide“ ??

Author Response

We would like to thank you for your careful reading, helpful comments, and constructive suggestions, which has significantly improved the presentation of our manuscript.

We have carefully considered all comments from the reviewers and revised our manuscript accordingly. In the following section, we summarize our responses to each comment from the reviewers. We believe that our responses have well addressed all concerns from the reviewers. We hope our revised manuscript can be accepted for publication.

  1. Page 2, line 53 and page 4, line 120: The order should be that the AR dimerizes first then translocates. The text lists the steps in a different order. Please change the order.

Response: We have corrected the AR dimerizes before transfer to nuclear. Accordingly, we have corrected the part involved in the article. 

  1. Line 664: Please correct these words:“enzlutrimide” must be corrected to “enzalutamide“

What is meant with „kalutrimide“ ??

Response: We have corrected the word “enzlutrimide”to “enzalutamide“. “Kalutrimide is a misspelling of our example AR antagonist. We have corrected it to “bicalutamide”.